# Gait Ability and Muscle Strength in Institutionalized Older Persons with and without Cognitive Decline and Association with Falls

**DOI:** 10.3390/ijerph182111543

**Published:** 2021-11-03

**Authors:** Maria dos Anjos Dixe, Carla Madeira, Silvia Alves, Maria Adriana Henriques, Cristina Lavareda Baixinho

**Affiliations:** 1Center for Innovative Care and Health Technology (ciTechcare), School of Health Sciences of Polytechnic of Leiria, Polytechnic of Leiria, 2410-541 Leiria, Portugal; manjos.dixe@gmail.com; 2Vila Franca Hospital, 2600-009 Vila Franca de Xira, Portugal; carla.madeira@hvfx.pt (C.M.); silvia.alves@hvfx.pt (S.A.); 3Nursing Research, Innovation and Development Centre of Lisbon (CIDNUR), 1600-190 Lisbon, Portugal; ahenriques@esel.pt

**Keywords:** falls, aging population, nursing home, gait, muscle strength

## Abstract

Falls are a complex problem, given their multifactorial nature, the comorbidities involved, and due to the dependency of older persons living in nursing homes. Risk, fear of falling, falls themselves, and their recurrence are the main factors behind fragility fractures, lack of independence, and increases in pain prevalence, and other comorbidities in older populations. The objectives of the present quantitative and longitudinal study were: (a) to characterize the cognitive state and fall frequency of older persons living in nursing homes; (b) to analyze the relationship between cognitive status and some fall risk factors; and (c) to associate cognitive decline, gait ability, and muscle strength of the examined institutionalized older persons with fall occurrence and recurrence over 12 months. The participants were 204 older persons who lived in Portuguese nursing homes, and data were collected from January 2019 to February 2020 by consulting medical records and applying the following instruments: the Mini-Mental State Examination, Timed Up and Go Test, and Medical Research Council Manual Muscle Testing Scale. Fall prevalence, assessed in two periods, 12 months apart, was similar in both samples (with and without cognitive decline) and close to 42%, and the annual recurrence rate was 38.3%. Older persons with no cognitive decline showed an association between gait speed and occurrence of first fall and recurrent fall (*p* < 0.05). Muscle strength and use of gait aid devices were not related to falls and their recurrence, regardless of mental state.

## 1. Introduction

As a public health problem, there is a high prevalence of falls in older persons. In addition to physical injuries, there are psychological consequences, such as fear of falling and social issues related to isolation and/or institutionalization [1,2]. However, after institutionalization, older persons are subject to higher fall risks, since fall prevalence in nursing homes is higher than that recorded in communities [1,2]. This situation is worsened by higher levels of dependency, higher incidence of chronic diseases, polymedication, and gait alterations [3,4]. After the first fall episode, recurrence is more frequent in this context, with a recurrence rate in the subsequent 12 months, ranging from 30% to 40% [5].

Another important aspect is that most institutionalized older persons have cognitive decline [5], and there is a relationship between this decline, gait alterations, fall risk, fall prevalence, and the severity of the injuries resulting from falls [5,6,7]. Executive function is the cognitive domain most commonly associated with gait dysfunction [6]. In one study, low performance in attention and executive function was associated with gait slowing, instability, and future falls [7]. Executive functions play an important role in fall risk—attention, sensory integration, and motor planning are sub-domains of executive function associated with the risk of falling due to gait dysfunction, while cognitive flexibility, judgment, and inhibitory control affect the risk of falling due to adopting risky behaviors [6].

The results of a study in Portugal indicated that 81.4% of the elderly without cognitive decline and 43.9% of those with cognitive decline who fell took longer than 12 s to perform the test. The test proved to be a good predictor of the occurrence of falls in both groups (with or without cognitive decline) [5].

Other studies have pointed out that, after the first fall, older persons tended to avoid activities that required them to be on their feet or walk, including showering, carrying out small household chores, walking in the external area of the house, and attending social events because of the injuries, pain, and fear of falling again [8]. Institutionalized older persons with cognitive decline were subject to the risk of having restrictions to mobility and activity imposed on them by their caretakers after the first fall episode, which promoted dependence and isolation [5]. Gait and balance alterations have been seen by caretakers as worrying risk factors that required special attention even in older persons without cognitive decline, which increased the chances of older persons having their mobility limited in nursing homes, especially in their external areas [5,9,10,11]. This impacted care delivery to this public [5] and caused caretakers to act in the place of older persons in many activities, which led to greater support in the execution of activities of daily living, dependence, gait ability loss, and muscle strength loss [5,8].

As one study showed [10], some older persons are institutionalized when they are still independent or with low levels of dependence. However, losing their daily routine and being liberated from the obligation of carrying out a series of everyday activities, together with decreases in confidence in their ability to execute these activities after a fall episode, contribute to increasing idleness and reducing physical fitness and, consequently, increasing the risk of falls, as well as morbidity and mortality.

A study with community-dwelling older persons concluded that people who practiced physical exercise for shorter times during the week and had lower muscle strength, especially women, were exposed to a higher fall risk [12]. Tests of the physical strength of both hands yielded results of 17.6 ± 8.0 kg in the recent fall group, which was lower than that in the non-fall group (20.7 ± 8.7 kg) [12].

Despite the recommendations that older adults should perform resistance exercises involving the major muscle groups at least twice a week, because strength training improves quality of life and prevents falls [13], reviewing the literature indicates that there are not many studies associating muscle strength with fall occurrence in nursing homes. A study published in 1994 showed that the relationship between muscle weakness and falls was probably modified by multiple characteristics of individuals, their cultures, and their environments [14]. A more recent study that aimed to evaluate the impact of inertial training on upper and lower extremity strength in older persons [13] concluded that the training group showed statistically significant percentage changes (from 37.1% to 69.1%) in pre- and post-training maximal force for all trained muscles, but did not associate muscle strength with fall risk and occurrence [13].

Another study, published in 1995, identified slow gait (odds ratio = 3.319 in logistic regression), changes in life conditions over the previous two years (odds ratio = 2.171), reductions in muscle strength of the quadriceps (odds ratio = 1.782), and eye disease (odds ratio = 1.897) as fall risk factors [15].

The review of the literature shows that there are few studies with institutionalized elderly people that explore falling and its recurrence with walking ability, muscle strength, and cognitive status.

The objectives of the present study were: (a) to characterize the cognitive state and fall frequency of older persons living in nursing homes; (b) to analyze the relationship among cognitive status and some fall risk factors: gender, gait, previous fall, consuming of ≥four medications or benzodiazepines, walking aid devices, stroke; and (c) to associate cognitive decline, gait ability, and muscle strength of the examined institutionalized older persons with fall occurrence and recurrence over 12 months.

## 2. Material and Methods

### 2.1. Study Design

A quantitative, correlational, and longitudinal study [16], with a 12-month interval between evaluations, aiming to associate cognitive decline, gait capacity, and muscle strength with fall occurrence and recurrence, with a 12-month interval, was conducted.

### 2.2. Participants

The study was carried out in two Portuguese nursing homes. The eligibility criteria for the participants were as follows: being 65 years old or older; being institutionalized; being independent regarding mobility (even if a gait aid device or wheelchair was used); and going through fall occurrence evaluations at two different times, 12 months apart. The sample is of the non-probabilistic type, and the sample size was not determined as data collection was carried out with all users of the selected institutions.

### 2.3. Data Collection

Data were collected between January 2019 and February 2020 by two nurses specializing in rehabilitation nursing, one from another institution, who were trained to apply the instruments and the fall evaluation.

The instrument used to determine cognitive decline was the Mini-Mental State Examination, European Portuguese version [17], which is scored according to the following criteria: ≤15 points for illiterate persons, ≤22 points for persons with up to 11 years of formal education, and ≤27 points for persons with more than 11 years of formal education. The higher values of the score indicate higher cognitive performance. It addresses questions referring to recent memory and immediate memory recall temporal and spatial orientation, attention and calculation, and language—aphasia, apraxia, and constructional ability [17].

Gait was evaluated by applying the Timed Up and Go Test, which allowed assessing gait ability, gait quality, and gait time. This simple and easy-to-apply test evaluates mobility ability and functional balance in older persons living in the community or in institutions [18]. The procedure was explained to the elderly individual before the test was performed. The elderly person was seated in a chair with lateral arm support and was asked to get up without leaning on the sides of the chair, walk 3 m, turn 180 degrees, and return to the starting point to sit down again. The test was performed only once for each participant at the beginning of the study.

Muscle strength was assessed by using the Medical Research Council Manual Muscle Testing Scale, which scores muscle strength from 0 to 5 [19]. For the purposes of the present study, the examined older persons were split into two groups. Group 1 included older persons with the ability to carry out movements against gravity (3: movement of the limb against gravity, but without resistance; 4: movement of the limb with at least some resistance; and 5: normal strength). Group 2 gathered the older persons who showed 0: absence of muscle contraction; 1: visible muscle contraction with or without sign of movement; and 2: movements of the limbs, but not against gravity. This option can be justified by the importance of antigravity muscles in gait quality, balance, and restoration of a stable position when a person accidentally trips or slips during gait.

The evaluation was made for each of the upper and lower limbs separately, because of the prevalence of stroke and other pathologies with interference in muscle strength. It should be noted that the inclusion criterion was that the person must be able to mobilize to perform the gait test.

Medical records were consulted to collect information about consumption of medications, diagnoses, and fall occurrence. The fall definition considered in the present study was that adopted by the World Health Organization based on code E880-E888 of the International Classification of Diseases, Ninth Revision. According to it, falls are all nonintentional events, resulting in a change of position to a lower level compared to the initial position, excluding intentional changes to rest on furniture, walls, or other objects [20]. The occurrence of falls was monitored and recorded by the nursing team in the medical record, where there is a specific field for this record.

### 2.4. Data Analysis

SPSS version 23.0 was used to statistically treat the data. The results were described using descriptive statistic parameters: absolute frequencies, measures of central tendency (means), and measures of dispersion and variability (standard deviation).

Before applying the statistical test on the relationship between variables, the Kolmogorov–Smirnov test was applied to assess the distribution of variables. Having found that the sample did not present a normal distribution, non-parametric techniques were used to test the relationship between the various variables under study [16].

The Mann–Whitney test, which is an alternative to the t-test for two independent samples, was applied. It compares the center of the location of the two samples to detect differences between the two corresponding populations. Additionally, the chi-square independence test was used to analyze the relationship between two qualitative variables. When the relationship between the variables was significant, the odd ratio was calculated with a 95% confidence interval, effect size was also calculated whenever the *p* value was <0.05 [16].

### 2.5. Ethical Considerations

The present study is part of a project about fall risk management in nursing homes. It was approved by the Universidade Católica Portuguesa Research Ethics Committee as per report no. 24/052013. Ethical principles related to consent, privacy, and confidentiality were observed during the development and execution of the study.

## 3. Results

### 3.1. Sociodemographic and Clinical Characteristics of the Older Persons by Cognitive State

The final sample was 204 older persons who met the inclusion criteria. Half the sample showed cognitive decline (26.5% were men and 73.5% were women). Among the residents who did not show cognitive decline, 31.4% were men and 68.6% were women. In both of these groups, older persons who were 85 years old or older prevailed (65.7% in the group without cognitive decline and 69.6% in the group with cognitive decline); no statistically significant differences were found (*p* > 0.05).

In the first period of evaluation, fall prevalence was high, because at least one fall was recorded during the previous year for 85 older persons (44 without cognitive decline and 41 with cognitive decline). It was found that 41.6% of the participants fell over the subsequent 12 months (40 without cognitive decline and 46 with cognitive decline), with 38.3% of the fall episodes being recurrent and 61.7% occurring in residents who had not fallen when the first evaluation was carried out.

For the variables that presented significance values we also calculated the odd ratio and it can be stated that users with cognitive disorders have a higher risk of taking benzodiazepines (2.962 (1.633–5.370)), of using a walking aid (2.976 (1.301–6.806)), and a lower risk/chance of having normal or lower TUGT values (0.271 (0.152–0.485)).

Table 1 shows the characteristics of the samples. There were statistically significant differences between having cognitive decline and not having cognitive decline only regarding gait self-care, consumption of benzodiazepines, and use of gait aid devices.

### 3.2. Muscle Strength, Gait, and Cognitive State of the Older Persons

Table 2 shows that there were statistically significant differences between the two cognitive states regarding only left upper limb muscle strength.

The older person with muscle strength in the left upper limb has a higher chance of belonging to the group of people without cognitive disorders (0.442 (0.216–0.904)).

Evaluation of gait speed, mobility capacity, and functional balance of the older persons as measured by the Timed Up and Go Test found no statistically significant differences when older persons with and without cognitive decline were compared (U = 1046.500; *p* > 0.05).

### 3.3. Gait, Muscle Strength, and Falls throughout the Year, According to Cognitive Decline of the Older Person

Table 3 indicates that muscle strength was not related to falls in the residents of nursing homes, regardless of presence of cognitive decline.

No statistically significant differences were found in muscle strength regarding occurrence of a fall episode over the year that followed the first fall (Table 4).

No statistically significant differences were found between gait speed, occurrence of the first fall episode, and a recurrent fall according to cognitive state, except for the older persons without cognitive decline who fell during the year: U = 557.000, (*p* < 0.05), as shown in Table 5. Regarding effect size, it was revealed to be small (effect size = 0.0609).

No statistically significant differences were found for the presence or absence of a fall episode regarding use of a gait aid device by the older persons with or without cognitive decline (Table 6).

## 4. Discussion

Fall prevalence in the two periods for which evaluations were carried out was about 46.1%, which agrees with results reported in other studies that recorded fall prevalence ranging from 13% [21] and 67% [22], and that noted that nursing homes showed recurrence rates and injury severity higher than those found in community-dwelling older persons, which led to serious consequences for functionality [5].

Falling was recurrent for 38.3% of the older persons. A literature review that had the objective of identifying the most commonly used instruments to determine fall risk in nursing homes concluded that professionals applied instruments ranging from risk assessment scales to functional evaluation tests, such as the Timed Up and Go Test and the Performance-Oriented Mobility Assessment (POMA), but that the question “Have you fallen over the past 12 months?” also had strong predictive value for occurrence of new episodes [2]. Given these data, association of prevalence with recurrence becomes a major risk factor for this population. In addition, fall secondary injuries, and their severity, pain, the postoperative immobility period that follows a fracture, and the fear of a new fall episode lead to restrictions in an older person’s activity and mobility, with gradual establishment of alterations related to immobility syndrome and their consequences [5,23,24].

Regarding gait speed evaluation, mobility capacity, and functional balance of the examined older persons assessed by means of the Timed Up and Go Test, no statistically significant differences were found when the older persons with and without cognitive decline were compared. There were significant differences between gait speed and occurrence of the first fall episode in people without cognitive decline. The results of another study indicated that lower gait speed was not significantly related to fall occurrence (*p* = 0.28) [25]. Studies in nursing homes have not been conclusive regarding an association between execution time in the Timed Up and Go Test and fall risk in this setting [2,25,26], in opposition to what was shown for community-dwelling older persons, in whom gait alterations were associated with fall risk and prevalence [27]. Gait speed has been associated with institutionalized health conditions and functional decline in older individuals [5,26].

There have been few studies associating gait speed and gait quality with falls in institutionalized older persons, and the few that have addressed this relationship have used different instruments to evaluate gait, which hinders the comparison between them. Therefore, the authors agreed that, to date, there is insufficient data to determine whether lower gait speed is associated with falls in nursing homes [25].

The issue of gait is sensitive in nursing homes, not only due to the dependence of institutionalized people, but also due to the way care is organized and provided, with little participation of the elderly in activities, and even with a certain lack of obligation in performing activities of daily living, which influences the loss of muscle strength and loss of gait quality. Future studies should explore the time spent performing the TUGT and gait quality in people with and without cognitive decline.

Considering this data scarcity and the lack of information on use of gait aid devices during the execution of the tests (it is known that a high percentage of older persons use a gait aid device or lean on sidebars during gait), future studies must explore the relationship between the time to execute the tests and the characteristics of gait aid devices, and the relationship between fall risk and prevalence and injury severity. Using mobility devices increases fall risk and injury severity in people 85 years old or older [28], but often provides these people with a level of independence that could not be reached without them. Every year, more than 47,000 older adults are treated in emergency departments in the United States because of falls related to mobility devices [28]. Most injuries resulting from the use of mobility devices involve walkers (87%) [28]. Although the results of the present study did not associate falls and their recurrence with the use of gait aid devices, future studies must investigate this association.

This study did not evaluate the strength of both hands and their relationship to falling. The high prevalence of elderly who use a walking aid warrants further studies to explore the relationship between hand strength, the use of walking, falls, and its recurrence. The results of one study observed that the greater the strength, in both hands, the lower the risk of falling; the strength of both hands tested was 17.6 ± 8.0 kg in the recent fall group, significantly weaker than that in the non-fall group (20.7 ± 8.7 kg) [12].

Gait, cognition, and falls are closely related [6,7]. This comorbidity and the interaction between gait anomaly and cognitive deficiency may be the basis of the high prevalence of falls in older adults with dementia [7]. Gait assessment and cognitive assessment, especially executive function, must be integrated into fall risk screening. The results of the evaluation must be interpreted and used by applying a multidisciplinary approach. Specific strategies such as customized training programs and behavioral change must be considered as part of a set of measures to prevent falls [6].

Most studies have associated falls with lower limb strength [24,29], and it has been well documented that both balance training and strength training have the potential to control risk factors inherent in falls in older persons [29,30]. Few studies have addressed the role played by upper limb strength, but older persons who show decreased gait ability end up resorting to gait aid devices. For these devices to be properly and safely used, older persons must have muscle strength in the upper limbs. Using gait aid devices is an important fall risk factor. Additionally, older persons who have already fallen have a greater chance of using gait aid devices [31], and users of these devices show higher probability of experiencing recurrent falls [3,6,31,32,33].

The association between use of gait aid devices and falls may be related to by gait alterations, incorrect use of the devices, and inadequacy of the chosen devices, given the anthropometric and biomechanical characteristics of the older persons [31,32,33,34]. Future studies should examine the relationship between muscle strength, correct use of gait aid devices, fall risk, fall occurrence, and fall recurrence.

### Study Limitations

Some limitations of the present study should be noted. Selecting a purposive sample can lead to biased results, which impairs their generalization. Therefore, a national study that includes nursing homes and takes into account their diversity is recommended.

Choosing a single instrument to assess gait and muscle strength may provide limited opportunities to identify other types of alterations and their association with cognitive decline and fall occurrence.

The data collection on falls was carried out by consulting the older persons’ clinical processes, which may have led to a bias by the hypothesis that unwitnessed falls or unrecorded falling episodes may have occurred.

## 5. Conclusions

Falls are a serious public health problem for institutionalized older persons, who are a more vulnerable population. The participants in the present study were institutionalized older persons in two Portuguese nursing homes, and 50% of the sample had cognitive decline. Fall prevalence, assessed in two periods, 12 months apart, was around 42% for both periods, a rate that can be considered high; the fall recurrence rate, recorded over a year, was 38.3%. In older persons without cognitive decline, there was an association among gait speed, occurrence of the first fall, and fall recurrence (*p* < 0.05). Muscle strength and use of gait aid devices were not related to falls or fall recurrence, regardless of cognitive state.

To assess the effectiveness of the introduction of fall prevention interventions, it is important to conduct an experimental or quasi-experimental study.

## Figures and Tables

**Table 1 ijerph-18-11543-t001:** Sociodemographic and clinical characteristics of older persons by cognitive state.

			Without Cognitive Decline		With Cognitive Decline		*X* ^2^	*p*
			N	%	N	%		
Age group	65 to 74 years		2	2.0	3	2.9	0.726 *	0.696
	75 to 84 years		33	32.4	28	27.5		
	85 years or older		67	65.7	71	69.6		
Gender	Male		32	31.4	27	26.5	0.382	0.537
	Female		70	68.6	75	73.5		
Gait performance (TUGT)	Time shorter than 12 s		38	37.3	70	68.6	18.909	0.000
	Time longer than 12 s		64	62.7	32	31.4		
Fall	No		58	56.9	61	59.8	0.081	0.776
	Yes		44	43.1	41	40.2		
≥four medications	No		7	6.9	8	7.8	0.000	1.000
	Yes		95	93.1	94	92.2		
Benzodiazepines	No		77	75.5	52	51.0	12.145	0.000
	Yes		25	24.5	50	49.0		
Gait aid devices	No		23	22.5	9	8.9	6.119	0.013
	Yes	Walking stick	16	15.7	4	4.0		
		Canadian crutches	23	22.5	11	10.9		
		Walker	17	16.7	14	13.9		
		Wheelchair	23	22.5	63	62.4		
Stroke	No		70	68.6	74	72.5	0.213	0.645
	Yes		32	31.4	28	27.5		
Fall over the next year	No		62	60.8	56	54.9	0.503	0.478
	Yes		40	39.2	46	45.1		

Legend: * two cells with expected values < 5.

**Table 2 ijerph-18-11543-t002:** Muscle strength and cognitive state of the older persons.

Muscle Strength	Group	With Cognitive Decline	Without Cognitive Decline	*X* ^2^	*p*
LULMS ^a^	1	88	75	4.396	0.036
	2	14	27		
RULMS ^b^	1	87	75	3.628	0.057
	2	15	27		
LLLMS ^c^	1	70	72	0.023	0.879
	2	32	30		
RLLMS ^d^	1	67	71	0.202	0.653
	2	35	31		

Legend: ^a^—left upper limb muscle strength; ^b^—right upper limb muscle strength; ^c^—left lower limb muscle strength; ^d^—right lower limb muscle strength.

**Table 3 ijerph-18-11543-t003:** Results of application of the chi-square test among muscle strength, falls throughout the year, and muscle strength, according to the cognitive status of the older persons.

Muscle Strength		Group	Did Not Fall	Fell	*X* ^2^	*p*
Without cognitive decline	LULMS ^a^	1	50	38	0.000	1.000
		2	8	6		
With cognitive decline	LULMS ^a^	1	45	30	0.005 ^c^	0.946
		2	16	11		
Without cognitive decline	RULMS ^b^	1	51	36	0.338	0.561
		2	7	8		
With cognitive decline	RULMS ^b^	1	45	30	0.005	0.946
		2	16	11		
Without cognitive decline	LLLMS ^c^	1	42	28	0.534	0.465
		2	16	16		
With cognitive decline	LLLMS ^c^	1	42	30	0.061	0.804
		2	19	11		
Without cognitive decline	RLLMS ^d^	1	41	26	1.023	0.312
		2	17	18		
With cognitive decline	RLLMS ^d^	1	42	29	0.000	1.000
		2	19	12		

Caption: ^a^—left upper limb muscle strength; ^b^—right upper limb muscle strength; ^c^—left lower limb muscle strength; ^d^—right lower limb muscle strength.

**Table 4 ijerph-18-11543-t004:** Results of the application of the chi-square test to muscle strength of the older persons regarding occurrence of a fall episode over the year that followed the first fall.

	Muscle Strength	Group	Did Not Fall	Fell	*X* ^2^	*p*
Without cognitive decline	LULMS ^a^	1	55	33	0.354	0.552
		2	7	7		
With cognitive decline	LULMS ^a^	1	39	36	0.572	0.450
		2	17	10		
Without cognitive decline	RULMS ^b^	1	54	33	0.125	0.724
		2	8	7		
With cognitive decline	RULMS ^b^	1	39	36	0.572	0.450
		2	17	10		
Without cognitive decline	LLLMS ^c^	1	44	26	0.173	0.678
		2	18	14		
With cognitive decline	LLLMS ^c^	1	39	33	0.000	0.990
		2	17	13		
Without cognitive decline	RLLMS ^d^	1	42	25	0.109	0.741
		2	20	15		
With cognitive decline	RLLMS ^d^	1	38	33	0.043	0.835
		2	18	13		

Caption: ^a^—left upper limb muscle strength; ^b^—right upper limb muscle strength; ^c^—left lower limb muscle strength; ^d^—right lower limb muscle strength.

**Table 5 ijerph-18-11543-t005:** Results of the application of the Mann–Whitney U test to gait speed regarding both the existence of a fall episode throughout the study year and the subsequent year, and the cognitive status of the older persons.

	Cognitive Status		N	Average of TUGT	U	*p*
Falls throughout the subsequent year	Without cognitive decline	No	47	35.99	563.500	0.059
		Yes	32	45.89		
	With cognitive decline	No	16	16.91	134.500	0.743
		Yes	18	18.03		
Falls throughout the study year	Without cognitive decline	No	41	34.59	557.000	0.029
		Yes	38	45.84		
	With cognitive decline	No	15	18.83	122.500	0.487
		Yes	19	16.45		

**Table 6 ijerph-18-11543-t006:** Results of the application of the chi-square test to use of gait aid devices regarding both the existence of a fall episode throughout the study year and the subsequent year, and the cognitive status of the older persons.

Fall Year	Cognitive Status	Gait Aid Device	Did Not Fall	Fell	Chi Square	*p*
Study year	Without cognitive decline	No	13	10	0.000	1.000
		Yes	45	34		
	With cognitive decline	No	5	4	0.000	1.000
		Yes	56	36		
Subsequent year	Without cognitive decline	No	17	6	1.495	0.221
		Yes	45	34		
	With cognitive decline	No	5	4	0.000	1.000
		Yes	50	42		

## Data Availability

Data are available only upon request to the authors.

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
