# Peer review of "Gait Ability and Muscle Strength in Institutionalized Older Persons with and without Cognitive Decline and Association with Falls"

_ijerph, 2021, doi:10.3390/ijerph182111543_

Round 1

Reviewer 1 Report

The study by dos Anjos Dixe and colleagues investigates possible predictors of falls in nursing home inhabitants. This is a very important topic for the care of elderly and the work investigates the relatiohsip of falss with cognitive ability, motor ability and strength.

The findings, that predictive factors were only found in the subjects without cognitive decline is important and the work will be a reference for future studies.

While I know that this might not be feasible, would it be possible to see if certain medications/drug groups are a potential preditor of falls in the study group. As many use multiple medications regulary this might be too complex.

Author Response

Response:

The study by dos Anjos Dixe and colleagues investigates possible predictors of falls in nursing home inhabitants. This is a very important topic for the care of elderly and the work investigates the relatiohsip of falss with cognitive ability, motor ability and strength.

The findings, that predictive factors were only found in the subjects without cognitive decline is important and the work will be a reference for future studies.

While I know that this might not be feasible, would it be possible to see if certain medications/drug groups are a potential preditor of falls in the study group. As many use multiple medications regulary this might be too complex.

In this study we didn’t explores all medications group and their influence in falling, we only associate it with benzodiazepines because previous studies, in Portugal, established this association.

Reviewer 2 Report

Dear authors,

I want to grant to authors for the effort made inn this data collection because this study cannot be considered as a scientific study attending to the design and methodology. In addition, the quality of data and information presented has got a very low quality. I have a lot of concerns about this study, but I´m going to indicate the worst mistakes that every one justify a rejection of this study:

  • The statistical treatment must be based on the aim of a scientific study. Regarding to this study, authors have compared participants with and without cognitive decline. However, the aims of the study were the next: “a) characterizing the cognitive state and fall frequency of older persons living in nursing homes; b) evaluating fall risk factors in older persons who attend nursing homes; and c) associating cognitive decline, gait ability, and muscle strength of the examined institutionalized older persons with fall occurrence and recurrence over 12 months”. It´s inconsistent objectives and statistical treatment used?
  • Didn´t author estimate the sample size for this study? What is you’re the statistical power of the results presented?
  • All studies must be replicated by any reader. It´s impossible to replicate the recruitment of participants followed in this study. However, it´s not specified nor protocol nor instrument used for assessing gait and muscle strength. This is a very important inconsistent of this manuscript.
  • Why didn´t author check normality of every variable used? There is not any reason attending to the sample size for using Mann-Whitney test in every pairwise comparison.
  • Why didn´t author reported odd ratio (with interval confidence) as complement of Chi-squared test and effect sizes as complement of t-test?
  • Tables 2-4 what is the meaning of 1 and 2?
  • Results section is horrible and present a very low quality. In fact, it´s lower than a master student. It´s necessary that authors study more statistical and read more before writing future studies.
  • Writing is very bad and is far or an acceptable scientific soundness. In this sense, authors have used 6 paragraphs with non-related studies in the introduction. They should summarize 6 paragraphs (lines 41-83) in an only paragraph where the results of the different studies will be related between them (for example).
  • A good discussion must be related the results found with other studies and explain the implications of the results as the mechanism of actions which explain them.

Author Response

Response:

I want to grant to authors for the effort made inn this data collection because this study cannot be considered as a scientific study attending to the design and methodology. In addition, the quality of data and information presented has got a very low quality. I have a lot of concerns about this study, but I´m going to indicate the worst mistakes that every one justify a rejection of this study:

  1. The statistical treatment must be based on the aim of a scientific study. Regarding to this study, authors have compared participants with and without cognitive decline. However, the aims of the study were the next: “a) characterizing the cognitive state and fall frequency of older persons living in nursing homes; b) evaluating fall risk factors in older persons who attend nursing homes; and c) associating cognitive decline, gait ability, and muscle strength of the examined institutionalized older persons with fall occurrence and recurrence over 12 months”. It´s inconsistent objectives and statistical treatment used?

We change the objective b) to analyse the relationship between cognitive status and some fall risk factors; increasing congruence with the research and with the article's content (page 1, line 11-12 and page2 line 90-91).

  1. Didn´t author estimate the sample size for this study? What is you’re the statistical power of the results presented?

Regarding to the sample size - The sample is of the non-probabilistic type, and the sample size was not determined due to the fact that data collection was carried out with all users of the selected institutions (page 3, line 104-106). 

This is one of the limitations of this study.

  1. All studies must be replicated by any reader. It´s impossible to replicate the recruitment of participants followed in this study. However, it´s not specified nor protocol nor instrument used for assessing gait and muscle strength. This is a very important inconsistent of this manuscript.

The gait was evaluated by the TUGT (page 3, lines 119-126) and muscle strength by the Medical Research Council Manual Muscle Testing Scale (page 3, lines 127-136).

  1. Why didn´t author check normality of every variable used? There is not any reason attending to the sample size for using Mann-Whitney test in every pairwise comparison.

We didn´t report that but before applying the statistical test on the relationship between variables, the Kolmogorov-Sminorv test was applied to assess the distribution of variables. Having found that the sample did not present a normal distribution, non-parametric techniques were used to test the relationship between the various variables under study (page 4, lines 153-156).

  1. Why didn´t author reported odd ratio (with interval confidence) as complement of Chi-squared test and effect sizes as complement of t-test?

The odd ratio was calculated, when the relationship between the variables was significant, with a 95% confidence interval, effect size was also calculated whenever the p value was < 0.05, (page 4, lines 161-163).

We introduced the values in the results (page 4, lines 183-186; page 5, lines 195-196).

  1. Tables 2-4 what is the meaning of 1 and 2?

In the tables 2-4 the nº 1 and 2 is related to the information given on the methods about -  Group 1 included older persons with the ability to carry out movements against gravity (3: movement of the limb against gravity, but without resistance; 4: movement of the limb with at least some resistance; and 5: normal strength), whereas Group 2 gathered the older persons who showed 0: absence of muscle contraction; 1: visible muscle contraction with or without sign of movement; and 2: movements of the limbs, but not against gravity (page 3, lines 127-136).

  1. Results section is horrible and present a very low quality. In fact, it´s lower than a master student. It´s necessary that authors study more statistical and read more before writing future studies.

We introduced more results (page 4, lines 183-166; page 5, lines 195-196).

  1. Writing is very bad and is far or an acceptable scientific soundness. In this sense, authors have used 6 paragraphs with non-related studies in the introduction. They should summarize 6 paragraphs (lines 41-83) in an only paragraph where the results of the different studies will be related between them (for example).

In the introduction we didn’t made much changes because the others’ reviews considered adequate for the type of study.

  1. A good discussion must be related the results found with other studies and explain the implications of the results as the mechanism of actions which explain them.

We explore the discussion a little further (page 7, line 267-272 and page 8, line 285-290).

Reviewer 3 Report

The present study sought to investigate fall prevalence and recurrence in older individuals living in nursing homes and relate fall events with cognitive decline, gait ability, muscle strength, and other comorbidities. The reported findings reveal a high prevalence of falls and fall recurrence in the cohort but a lack of association between the risk of falling, cognitive decline, and muscle strength. Overall, the manuscript is nicely written, addressing an important topic for the health care of institutionalized older people. However, there are details of the methods requiring a better description. Some parts of the results section and accompanying tables should also be revised to improve understanding. The discussion should be improved.

Abstract

1 - The abstract is nicely written and informative. However, it contains no information about the association between cognitive decline and fall prevalence. The authors could add a short note to the abstract saying the fall prevalence was similar in the with- and without-cognitive decline participants.

Introduction

2 - The introduction offers a good account of the study's background and provides a clear definition of the study's aims and objectives. There are some minor issues that the authors could take into consideration.

  1. a) Page 1, line 39. In this sentence, the meaning of "gait dysfunction" is not apparent. Please, rephrase the sentence so one can better understand its content.
  2. b) Page 1, lines 40-44. At the end of the sentence, we read "... with the differences showing statistical significance in both groups." However, after reading this sentence we end up not knowing what are the differences showing statistical significance. Could it be the difference in TUG test performance between fallers and non-fallers in cognitive decline and non-decline groups? Would you please rewrite the sentence to make this point clear?
  3. c) Preceding the last paragraph, a summary of the rationale for conducting the study is missing. After reviewing the literature, the authors could/should identify the knowledge gaps that justify the need to carry out the present study before announcing the aim and objectives of the research.

Materials and methods

3 - Page 3, lines 103-106. This paragraph describes the scoring system for the Mini-Mental State Examination, indicating the cutoff points according to different educational levels. I assume these are cutoffs for the participants being considered as manifesting cognitive decline or not, but the sentence has to be rewritten to make this understanding clear. Also, the use of the names "sick residents", seemingly to refer to the participants, leads to confusion. Please, consider replacing "sick residents" with "participants" or other designation used in the manuscript.

4 - The manuscript lacks a description of the TUG test procedures and the manual muscle strength testing. For the TUG test, indication of the walking distance, starting and ending posture, number of repetitions and instructions given to the participants, should be reported. For muscle testing, it the descriptions of the tested muscle groups or movements and the body position of the tests should be added to the manuscript. The authors claim the participants were separated into two groups based on whether they were able to contract the muscles and move the limbs against gravity or not. It is difficult to assess the significance of muscle testing outcomes without knowing which muscles were tested. However, it is unlikely that participants able to walk and perform the TUG test are not capable of contracting limb muscles and move their upper and lower limbs against gravity. Could the authors provide information regarding their manual muscle testing reliability (test-retest and inter-rater reliability) and explain how participants with a score below 3 are capable of walking and perform the TUG test?

5 - Fall occurrence data were obtained through medical records. Were there any procedures implemented to confirm the medical records data?

6 - Please explain the reason for using the non-parametric Mann-Whitney test instead of the parametric t-test and specify which variables were analyzed with the Mann-Whitney test.

Results

7 - Page 4, line 5. A p value (non-significant) is reported in this sentence. I assume this p-value relates to chi-square test (please also explain why in Table 1 there is no chi-square value for age group frequencies). I suggest the authors report the chi-square statistic, including the degrees of freedom, sample size, chi-square value, and the significance level.

8 - Table 1. What is the meaning of "gait self-care"? Does this term refer to two subgroups separated based on TUG test performance? It is also puzzling that participants with cognitive decline reveal faster gait speed and less reliance on walking aids. Also, there are many wheelchair users in the cohort, particularly in the with-cognitive decline subgroup. Do these participants depend on the wheelchair for locomotion, although they can walk and complete the TUG test (one of the eligibility criteria)?

9 - Page 5, line 187. Please, report the Mann-Whitney U value together with the respective p-value.

Discussion

Overall, the discussion is nicely written. However, it should be more focused on the study's findings instead of reviewing the literature. For instance, the results of the TUG test and their relation with fall events deserve a more in-depth discussion. The discussion should explain the better TUG performance by participants with cognitive decline than participants without cognitive decline. TUG performance is predictive of falls in community-dwelling elders but not in institutionalized elders, together with the highly protracted gait speed found in the present study (M = 45.89 sec, Table 5), raises concerns about the use of this test in this population.

Another key finding to be more thoroughly discussed is the lack of a relationship between muscle strength and fall prevalence. In independent community-dwelling elders, muscle strength and, even more importantly, muscle power of lower limb muscles is a decisive fall risk factor. Since such a relationship could not be proved by the present study, combined with the severe frailty of the cohort, as judged by the reported muscle strength and TUG tests, suggest other factors determine the fall risk in this group. Likely reasons are lack of mobility, rare opportunities for unassisted walking, and the daily routine followed by the nursing home residents. These reasons should be discussed and means to control for their effect on the assessment of risk fall proposed.

Other minor issues

Table 3, caption. "Level of consciousness" is not appropriate to designate cognitive ability status. Would you please replace this term with a more accurate one?

Table 5. The column heading "Average" is confusing. I suppose the respective column reports the mean time taken to complete the TUG test, and the column heading should give this information together with the measuring units (seconds).

Table 6. Please, change the column heading "Level of consciousness". 

Author Response

Response:

The present study sought to investigate fall prevalence and recurrence in older individuals living in nursing homes and relate fall events with cognitive decline, gait ability, muscle strength, and other comorbidities. The reported findings reveal a high prevalence of falls and fall recurrence in the cohort but a lack of association between the risk of falling, cognitive decline, and muscle strength. Overall, the manuscript is nicely written, addressing an important topic for the health care of institutionalized older people. However, there are details of the methods requiring a better description. Some parts of the results section and accompanying tables should also be revised to improve understanding. The discussion should be improved.

Abstract

1.The abstract is nicely written and informative. However, it contains no information about the association between cognitive decline and fall prevalence. The authors could add a short note to the abstract saying the fall prevalence was similar in the with- and without-cognitive decline participants.

We made the suggested change (page 1, line 18).

Introduction

The introduction offers a good account of the study's background and provides a clear definition of the study's aims and objectives. There are some minor issues that the authors could take into consideration.

  1. Page 1, line 39. In this sentence, the meaning of "gait dysfunction" is not apparent. Please, rephrase the sentence so one can better understand its content.

We rephrased the sentence (page 1, line 38-41).

  1. Page 1, lines 40-44. At the end of the sentence, we read "... with the differences showing statistical significance in both groups." However, after reading this sentence we end up not knowing what are the differences showing statistical significance. Could it be the difference in TUG test performance between fallers and non-fallers in cognitive decline and non-decline groups? Would you please rewrite the sentence to make this point clear?

We rephrased the sentence (page 2, line 46).

  1. Preceding the last paragraph, a summary of the rationale for conducting the study is missing. After reviewing the literature, the authors could/should identify the knowledge gaps that justify the need to carry out the present study before announcing the aim and objectives of the research.

The review of the literature showed that there are few studies with institutionalized elderly people that explore falling and its recurrence with walking ability, muscle strength and cognitive status and this is the justification for the conduction of this study. The information about this gap were introduced before the objective of this study (page 2, line 86-88).

Materials and methods

  1. Page 3, lines 103-106. This paragraph describes the scoring system for the Mini-Mental State Examination, indicating the cutoff points according to different educational levels. I assume these are cutoffs for the participants being considered as manifesting cognitive decline or not, but the sentence has to be rewritten to make this understanding clear. Also, the use of the names "sick residents", seemingly to refer to the participants, leads to confusion. Please, consider replacing "sick residents" with "participants" or other designation used in the manuscript.

The term sick was replaced by person. In regarding to the scoring system the higher values of the score indicate higher cognitive performance (page 3, line 113-115).

The test addresses questions referring to recent memory and immediate memory recall temporal and spatial orientation, attention and calculation, and language - aphasia, apraxia and constructional ability (page 3, line 115-118)..

  1. The manuscript lacks a description of the TUG test procedures and the manual muscle strength testing. For the TUG test, indication of the walking distance, starting and ending posture, number of repetitions and instructions given to the participants, should be reported. For muscle testing, it the descriptions of the tested muscle groups or movements and the body position of the tests should be added to the manuscript. The authors claim the participants were separated into two groups based on whether they were able to contract the muscles and move the limbs against gravity or not. It is difficult to assess the significance of muscle testing outcomes without knowing which muscles were tested. However, it is unlikely that participants able to walk and perform the TUG test are not capable of contracting limb muscles and move their upper and lower limbs against gravity. Could the authors provide information regarding their manual muscle testing reliability (test-retest and inter-rater reliability) and explain how participants with a score below 3 are capable of walking and perform the TUG test?

For the execution of the TUGT the procedure was explained to the older person before the test was performed. The elderly was seated in a chair with lateral arm support and was asked to get up without leaning on the sides of the chair, walk 3 meters, turning 180 degrees, and return to the starting point to sit down again. The test was performed only once for each participant at the beginning of the study (page 3, line 122-126).

Regarding to the commentary ‘it is unlikely that participants able to walk and perform TUGT are not capable of contracting limb muscles and move their upper and lower limbs against gravity’ - The evaluation was done for each member separately. People with hemiplegia have a score of 0 in the hemiplegic part of the body, but can have maintained strength in the contralateral hemibody and have the ability to walk and perform the TUGT.

  1. Fall occurrence data were obtained through medical records. Were there any procedures implemented to confirm the medical records data?

The occurrence of falls was monitored and recorded by the nursing team in the medical record, where there is a specific field for this record (page 3, line 147-148). One of the researchers trained the whole team on communication and reporting of falls in both institutions.

  1. Please explain the reason for using the non-parametric Mann-Whitney test instead of the parametric t-test and specify which variables were analyzed with the Mann-Whitney test.

The options for non-parametric test is justified because before applying the statistical test on the relationship between variables, the Kolmogorov-Sminorv test was applied to assess the distribution of variables. Having found that the sample did not present a normal distribution, non-parametric techniques were used to test the relationship between the various variables under study (page 4, line 153-156).

Also in some variables not all groups have n equal or greater than 30 to apply the central limit theorem.

Results

  1. Page 4, line 5. A p value (non-significant) is reported in this sentence. I assume this p-value relates to chi-square test (please also explain why in Table 1 there is no chi-square value for age group frequencies). I suggest the authors report the chi-square statistic, including the degrees of freedom, sample size, chi-square value, and the significance level.

When the relationship between the variables was significant, the odd ratio was calculated with a 95% confidence interval, effect size was also calculated whenever the p value was < 0.05 (page 4, line 161-163).

  1. Table 1. What is the meaning of "gait self-care"? Does this term refer to two subgroups separated based on TUG test performance? It is also puzzling that participants with cognitive decline reveal faster gait speed and less reliance on walking aids. Also, there are many wheelchair users in the cohort, particularly in the with-cognitive decline subgroup. Do these participants depend on the wheelchair for locomotion, although they can walk and complete the TUG test (one of the eligibility criteria)?

Gait self-care refers to the independent realization of the TUGT, we change the designation to TUGT performance. The elderly who use a wheelchair to get around and were included in the study have diminished walking ability, with higher scores on the TUGT, but were able to perform the procedure. Some, due to balance alterations and/or previous falls, end up being 'placed' in wheelchairs by the teams to prevent falling (page 4, table 1).

  1. Page 5, line 187. Please, report the Mann-Whitney U value together with the respective p-value.

The value is U= 557.000, it was introduced in the text (page 6, line 120).

Discussion

Overall, the discussion is nicely written. However, it should be more focused on the study's findings instead of reviewing the literature. For instance, the results of the TUG test and their relation with fall events deserve a more in-depth discussion. The discussion should explain the better TUG performance by participants with cognitive decline than participants without cognitive decline. TUG performance is predictive of falls in community-dwelling elders but not in institutionalized elders, together with the highly protracted gait speed found in the present study (M = 45.89 sec, Table 5), raises concerns about the use of this test in this population.

Another key finding to be more thoroughly discussed is the lack of a relationship between muscle strength and fall prevalence. In independent community-dwelling elders, muscle strength and, even more importantly, muscle power of lower limb muscles is a decisive fall risk factor. Since such a relationship could not be proved by the present study, combined with the severe frailty of the cohort, as judged by the reported muscle strength and TUG tests, suggest other factors determine the fall risk in this group. Likely reasons are lack of mobility, rare opportunities for unassisted walking, and the daily routine followed by the nursing home residents. These reasons should be discussed and means to control for their effect on the assessment of risk fall proposed.

we deepened the discussion (page 7, line 267-272 and page 8, line 285-290).

Other minor issues

  1. Table 3, caption. "Level of consciousness" is not appropriate to designate cognitive ability status. Would you please replace this term with a more accurate one?

We replace the designation (page 5, line 209).

  1. Table 5. The column heading "Average" is confusing. I suppose the respective column reports the mean time taken to complete the TUG test, and the column heading should give this information together with the measuring units (seconds).

It refers to the average time in the execution of the TUGT (page 6, table 5).

  1. Table 6. Please, change the column heading "Level of consciousness". 

Changed to ‘cognitive status’ (page 6, table 6).

Round 2

Reviewer 1 Report

Thank you for the opporunity to review this work.

Author Response

Dear review:

Thank you very much for your attention and time spent reviewing our article.

Best regards;

Reviewer 2 Report

Dear authors,

I´m very unhappy to check that authors after all the important concerns about the manuscript, authors haven´t enhanced the quality of their manuscript. First of all, I´m concerned after reading this answer “In the introduction we didn’t made much changes because the others’ reviews considered adequate for the type of study” and this other “We explore the discussion a little further”. I must say you that the writing of this manuscript is one of the worst that I have ever read. In fact, it´s insufficient for exceeding the minimum in my master student.

The introduction of the manuscript is a bad narrative review about this topic. Authors have selected randomly some studies that have been summarized and not related. Authors must know that an introduction must contain a relationship between the variables of a study. If you read your introduction, you will check that the introduction is wrong.

The same is observed in the discussion section where it hasn´t included information related to the mechanism of action that explain a good or not association with cognitive status. All the discussion section is insufficient and it´s wrong.

I´m surprised with the quickly changes of the aims of the study. However, authors want “to characterize the cognitive state and fall frequency of older persons living in nursing homes”. At the same time, authors said to me that it´s not necessary to estimate sample size. I´m very surprised because for making a characterization, it´s necessary to estimate sample size for obtaining a statistical power. In fact, based on the participants recruitment, this objective isn´t possible because “the users of a selected institution” is not representative and it exists a lot of variables that could affect to the results reported.

The conceptualization of the objective of this study is wrong! And I´m very worried because authors didn´t include a limitations of study section including the limitations indicated.

Regarding to the method section, this study is not reproductible. Reader cannot know the equipment use for assessing the time in TUG, for example. I´m not sure that the results reported could be reliable.

Regarding to the statistical treatment, there isn´t exist any little of sense. Could authors report that this sample size isn´t a normal distribution. If it is true, I request the results of this analysis because I cannot believe it. None variable is attending to a normal distribution?? I´m really concerned.

Author Response

Dear Review:

We thank you, once again, for your detailed and careful review of our article.
Regarding the methodological questions you raise:

This study has limitations related to the type and size of the sample and, therefore, the results only result from data collection in two Portuguese nursing homes. Therefore, a national study that includes nursing homes and takes into account their diversity is recommended.

With regard to data collection on falls among the elderly, it was carried out by consulting the elderly's files by IRPI professionals and not by the researchers, and by direct observation. This may have been biased because not all falls that occurred during the study were recorded.

To assess the effectiveness of the introduction of fall prevention interventions, it is important to conduct an experimental or quasi-experimental study.

For all variables it was determined whether the distribution was normal, through the Kolmogorov- Smirnov test and the homogeneity of variances through Levene's test. For all variables, without exception, the p-value is less than 0.001 guaranteeing non-normality.  For the variable relationships in table 1, 2, 3, 4 and 6 we have qualitative variables so the appropriate test is the Chi square test. Next we present the Kolmogorov- Smirnov test values for the relationship between the variables of Fall over the study year (KS=0.343; p=0.000) and Fall over the subsequent year (KS= 0.370; P= 0.000) and Average of TUGT (KS=0.102; p=0.006). For the use of parametric tests (Student's t) we did not meet the conditions for the use of the central limit theorem because in some groups we did not have at least 30 individuals.

We hope we have been able to answer your concerns.

Best regards;